# Grainyhead-like (Grhl) Target Genes in Development and Cancer

**DOI:** 10.3390/ijms23052735

**Published:** 2022-03-01

**Authors:** Jemma G. Gasperoni, Jarrad N. Fuller, Charbel Darido, Tomasz Wilanowski, Sebastian Dworkin

**Affiliations:** 1Department of Physiology, Anatomy and Microbiology, La Trobe University, Melbourne, VIC 3086, Australia; jemma.gasperoni@latrobe.edu.au (J.G.G.); jarrad.fuller@latrobe.edu.au (J.N.F.); 2The Peter MacCallum Cancer Centre, 305 Grattan St, Melbourne, VIC 3000, Australia; charbel.darido@petermac.org; 3Sir Peter MacCallum Department of Oncology, The University of Melbourne, Parkville, VIC 3010, Australia; 4Institute of Genetics and Biotechnology, Faculty of Biology, University of Warsaw, 02-096 Warsaw, Poland; t.wilanowski@biol.uw.edu.pl

**Keywords:** transcription factor, grainyhead-like, target genes, development, cancer

## Abstract

Grainyhead-like (GRHL) factors are essential, highly conserved transcription factors (TFs) that regulate processes common to both natural cellular behaviours during embryogenesis, and de-regulation of growth and survival pathways in cancer. Serving to drive the transcription, and therefore activation of multiple co-ordinating pathways, the three GRHL family members (GRHL1-3) are a critical conduit for modulating the molecular landscape that guides cellular decision-making processes during proliferation, epithelial-mesenchymal transition (EMT) and migration. Animal models and in vitro approaches harbouring GRHL loss or gain-of-function are key research tools to understanding gene function, which gives confidence that resultant phenotypes and cellular behaviours may be translatable to humans. Critically, identifying and characterising the target genes to which these factors bind is also essential, as they allow us to discover and understand novel genetic pathways that could ultimately be used as targets for disease diagnosis, drug discovery and therapeutic strategies. GRHL1-3 and their transcriptional targets have been shown to drive comparable cellular processes in Drosophila, C. elegans, zebrafish and mice, and have recently also been implicated in the aetiology and/or progression of a number of human congenital disorders and cancers of epithelial origin. In this review, we will summarise the state of knowledge pertaining to the role of the GRHL family target genes in both development and cancer, primarily through understanding the genetic pathways transcriptionally regulated by these factors across disparate disease contexts.

## 1. Introduction

The best animal models are those that closely recapitulate key aspects of a disease in humans. By extension, genes that perform the same biological role across multiple species (i.e., are “highly conserved”) are of strong interest to the research community, as it is very likely that the key roles played by these genes in lower organisms will also be performed in humans. Moreover, their expected transcriptional regulatory networks will also be highly conserved across species, thereby allowing for rapid insight into disease-causing mechanisms that underpin both developmental (congenital) and adult-onset disease (e.g., cancer) in humans.

One such family of genes, namely the “Grainyhead-like” (Grhl) family of transcription factors (TFs) fits these criteria. First discovered and named in Drosophila [1], orthologues of this gene have been identified and experimentally characterised in mice [2,3], C. elegans [4], zebrafish [5,6,7], fungi [8], the insect subphylum Hexapoda [9] and humans [10,11,12]. Although comprising only a single gene in insects and C. elegans, vertebrates harbour three orthologues (generally termed Grhl1-3), with the zebrafish genome duplication resulting in a further sub-functionalisation of grhl2 into grhl2a and grhl2b [13]. 

Notably, this gene family plays a multitude of roles during normal embryogenesis, as evidenced through the generation of numerous animal gene-deletion models [14]. Regulating processes such as closure of the neural tube [3,15], morphogenesis and cellular survival of the brain [13,16], numerous aspects of craniofacial development and secondary palate fusion [7,12,17,18], formation of the lungs [19], and establishment and maintenance of a functional skin barrier [20,21], this gene family is integral for normal embryonic development. Moreover, the grh/Grhl factors have recently been characterised as “pioneer TFs” [22,23,24], that is, they are able to access DNA-binding sites within typically inaccessible (closed) chromatin, to facilitate direct target gene transcription, further solidifying the essential nature of these TFs in normal cellular function and as key drivers of embryogenesis.

However, as with many genes that are indispensable for embryogenesis, the GRHL factors are often also deregulated in adult diseases, typically cancer, that are characterised by cellular de-differentiation or loss of functional integrity [25]. GRHL3 is a potent tumour-suppressor gene [26], and is implicated in numerous cancers of epithelial origin, such as head and neck squamous cell carcinoma [27], breast [28] and skin [29,30]. GRHL2 is also known to function in cancer, both as an oncogene [31,32,33,34,35], but also as a potential tumour-suppressor gene, through inhibition of the epithelial-mesenchymal transition (EMT) [36,37,38]. Lastly, GRHL1 appears to have a more minor role in tumorigenesis, nonetheless it is implicated as a tumour suppressor gene in neuroblastoma [39] and oesophageal cancer [40] and loss of Grhl1 predisposes animals to tumour development in a skin cancer model [41]. Ultimately, the GRHL factors act as molecular conduits to preserve structural integrity and regulate proliferation—key hallmarks of both embryogenesis and cancer progression.

These disparate roles played by the GRHL family make the identification of target genes and transcriptional mechanisms critical, as a greater understanding of the molecular pathways by which GRHL-genes exert their functions raises the exciting possibility that novel druggable targets may be discovered to ameliorate the severity of both congenital defects and epithelial cancers. 

Although this TF family directly regulates multiple processes common to both development and cancer, what is becoming very clear is that the regulatory specificity of these genes is highly context-dependent [42]. A nice analysis of GRHL2-function in the context of the EMT has recently been published [43], and so we will not cover this role again here; however, clearly the role of Grhl-factors in maintaining epithelial integrity and stability, and preventing mesenchymal transition is essential. 

Nonetheless, this review will identify some of the experimentally validated target genes and downstream pathways governed by GRHL-signalling across tissue-specific development, cancer, and where appropriate, non-cancer epithelial diseases, and highlight the specific processes in which particular genetic networks are activated. A summary of the organs and organisms in which the target genes have been identified is presented in Figure 1, Figure 2, Figure 3 and Figure 4 and Table 1, Table 2, Table 3 and Table 4, that highlight the specific model organism or cell line in which these functional interactions were discovered and characterised. Unless otherwise indicated within the tables, “direct targets” of grh/Grhl factors are those for which there exists evidence of functional regulation (e.g., through specific activation assays in vitro or in vivo, or changes in gene expression following grh/Grhl dysregulation), together with clear evidence that grh/Grhl binds to the gene promoter (e.g., via ChIP or ChIP-Seq assays).

## 2. Neural Tube Closure and Morphogenesis

During neurulation, neural tube fusion progresses from the dorsal midline, resulting in the neuropores in the brain and spine eventually closing [102]. The coordination of tissues within the neural ectoderm and surface ectoderm mediates the bending of the neural plate which folds up and “zippers” along the dorsal midline. The bending of the neural plate occurs at the median hinge point (MHP) that surrounds the notochord, and the paired dorsal-lateral hinge points (DLHP), that arise in the lateral neural folds [103]. 

Although Grhl2 and Grhl3, albeit not Grhl1, are important for proper neural tube closure, they have distinct expression patterns. Around the time of neural tube closure in mice, Grhl2 is expressed throughout the entire non-neural ectoderm (NNE). By comparison, Grhl3 initially shows restricted expression in only the surface NNE, directly adjacent to the folding neural plate [3] and subsequent expression is seen in spinal neuroepithelium and hindgut endoderm at E10-10.5 [104]. These expression data suggest disparate roles in neurulation.

Notably, Grhl2 and Grhl3 have both unique and overlapping functions in neural tube closure, and critically, they transactivate distinct target genes in the neural tube. Grhl2 gene-deletion (Grhl2^−/−^) mouse models have been invaluable for delineating a role for Grhl2 in neural tube closure. Grhl2^−/−^ mice show fully penetrant exencephaly, alongside both rostral and caudal tube closure defects [105]. To date, at least five different Grhl2^−/−^ mouse models have been described in the literature (reviewed in [14]). Whereas three of these models result in embryonic lethality by E11.5 [15,105,106] and present with near-identical phenotypes (failed neural tube closure resulting in thoraco-lumbo-sacral spina bifida), two other models result in some embryos “escaping” to E18.5 and presenting with complete exencephaly and cleft face [16,107], alongside forebrain defects due to apoptosis [18]. An important finding from these studies is that the genetic background of the mice substantially influences phenotypic penetrance. Analysis of the Grhl2clf3 model found that 100% of Grhl2clft3 mice with an FVB/NJ background presented with reduced telencephalon size, whereas none of the mice on an A/J background presented with this phenotype [18]. Similarly, a separate study reported 100% mortality of Grhl2Gt(AC0205)^Wtsi^ mice at E9.5 on a C57BL/6 compared to BALB/c background; the latter of which survived until E10.5 and E11.5 [108], indicating that disparate gene expression is likely to exacerbate and influence Grhl2 function.

The analysis of differentially regulated target genes in these models has unearthed novel likely targets that are effectors of Grhl2 signalling, as well as shedding light on the molecular pathways and processes that are regulated. Analysis of gene expression by qRT-PCR in the neural folds of Grhl21Nisw mutants revealed that 25% of the most highly-downregulated genes encoded proteins important for adhesion, including E-cadherin, cadherin 3, Desmoglein 2 and Desmocollin 2, alongside genes from the Claudin family, namely Claudin 4/6/7 [109].

Similarly, loss of Grhl2 in an ENU mutagenesis mouse model showed disturbed epithelial integrity in the NNE. Histological analysis showed that preceding joining, the folds in both the forebrain and hindbrain of Grhl2 mutants did not have the normal tight cell-cell contact in the squamous cell layer of the NNE [110]. The NNE cells in the Grhl2 mutants displayed a shift towards mesenchymal characteristics, with higher expression of the cytoskeletal protein and mesenchymal marker Vimentin, and increased cellular motility dynamics, as shown by live imaging data. 

Notably, this was the first study to characterise the transcriptome via RNA-Seq of Grhl2-/- mice and identified that five GRHL2-regulated genes, (Sostdc1, Tmprss2, Esrp1, Fermt1, Lamc2), essential for EMT suppression in the NNE, were downregulated in Grhl2-/- embryos [108]. By promoting epithelial fate, GRHL2 can maintain the epithelial properties of cells in the NNE which is essential for neural tube closure. Furthermore, the downregulation of epithelial genes such as E-cadherin and the abnormal N-cadherin expression identified in the surface ectoderm layer are consistent with this finding [107,109,110]. 

Grhl2 overexpression has also been implicated in the aetiology of the spinal-cord closure defect spina bifida. Spina bifida occurs at a high frequency in axial defects (Axd) mutant mice that present with significantly up-regulated Grhl2 expression [106]. Collectively, these findings demonstrate that both over-expression and knock-out models of Grhl2 result in defects of neural tube closure, suggesting that mis-regulation of key Grhl2 downstream targets is responsible for the fusion defects seen in these models. 

Grhl3 is also important for proper neural tube closure. Two separate murine models of Grhl3-deletion [3,111] present with convergence-extension defects, open neural tube, fully-penetrant thoraco-lumbo-sacral spina bifida and sporadic exencephaly, indicative of failed neurulation. Moreover, a naturally-occurring, spontaneous model of spina bifida that has been used since the mid-1950s, “curly-tail” (*ct*; reviewed in [112]) has been linked to a point-mutation in an upstream, putative enhancer region of Grhl3 [104]. Similarities between *ct* mice and Grhl3 mutants include the chromosome on which the mutation is located in *ct* mice, and the fully penetrant spina bifida phenotype seen in Grhl3 mutants [3,111]. Furthermore, over-expression of Grhl3 driven by BAC transgenesis showed complete rescue of spina bifida in *ct* mice [106], confirming the functional relationship between Grhl3 and the curly-tail locus.

Unlike the majority of genes that are implicated in the aetiology of neural tube defects in mice, but not in humans, genetic variations in human GRHL3 do indeed present with numerous congenital defects, particularly in craniofacial development (reviewed in [16]), but also in the aetiology of spina bifida, highlighted by a study that identified eight deleterious variants in GRHL3 in a large sequencing study of human spina bifida cases [113]. Similarly, a separate study found that a GRHL3 missense mutation increased the risk of spina bifida and encephalocele phenotypes [114], and lastly, DNA methylation analyses revealed that hypomethylation of GRHL3 is associated with neural tube defects [115]. Taken together, these studies speak to the key, cross-species functional conservation of Grhl3-signalling in neurulation.

Mechanistic studies have suggested that defects in Planar Cell Polarity (PCP) [75], insufficient cell proliferation in the hindgut endoderm [116], and abnormal cell function in the non-neural surface ectoderm [116] may underpin defects of Grhl3-mediated neurulation, and hence, like in Grhl2^−/−^ embryos, it is likely that multiple genetic mechanisms driven by disparate target genes underpin these processes. Importantly, both Grhl2 and Grhl3 are expressed in the non-neural surface ectoderm and not the neuroepithelium at the time of neural tube closure. Grhl2 is expressed throughout the entire non-neural surface ectoderm whereas Grhl3 is predominately expressed in the region of the non-neural surface ectoderm directly adjacent to the neuroepithelium, an area which will ultimately overlay the neural tube [116].

During neural tube closure, canonical Wnt signalling and its antagonists DKK1/KREMEN temporally and spatially control the specification of progenitor cells into the neuroepithelium or SE fate [117]. The balance in cell fate specification between NE and SE cells within the neural plate border just prior to neurulation is likely crucial for the midline fusion propagation of the neural folds and separation of the neural tube and SE. This study also found that Grhl3 was acting downstream of Wnt and was responsible for SE specification of uncommitted progenitors within the neural fold [118]. At the time of epidermal differentiation, GRHL3 is localised to the cytoplasm and cell membrane [117]. The cytoplasmic localisation of GRHL3 appears to be important for the subsequent expression of PCP components VANGL Planar Cell Polarity Protein 2 (VANGL2) and Cadherin EGF LAG Seven-Pass G-Type Receptor 1 (CELSR1) [119]. Similarly, studies in vivo found the expression of Ras Homolog Family Member A (RhoA), and CELSR1 were downregulated in Grhl3cre/cre mice. In contrast, there were marked increases in the expression of these genes in Grhl3 transgenic embryos, indicating that Grhl3 is upstream of these factors.

In addition to the up-regulation in PCP signalling, actomyosin networks are also crucial during this time. Transient inhibition of Grhl3-function in keratinocytes in vitro shows that Grhl3 is important for actin enrichment [75]. ChIP and gene expression analysis of Grhl3^−/−^ mice identified ARHGEF19 as a direct target of GRHL3 [75], and this mechanism was integral for driving actin cytoskeleton remodelling. In similar findings in-vivo, Grhl3^cre/cre^ mutants present with less actomyosin at the junction of NNE and neural folds along the rostral-caudal axis [120]. This suggests that up-regulation of PCP signalling followed by direct transactivation of ARHGEF19 to driving acto-myosin enrichment is crucial for the subsequent biomechanical contractility required for complete neural tube closure. Lastly, differences in closure along the neural tube axis in Grhl3-mutants may be explained by the spatial differences in lamellipodia and filipodia exhibited by Grh3^−/−^ embryos [119]. Electron microscopy revealed that Grhl3 mutants have fewer lamellipodia in the caudal region during neural tube closure and increased filipodia at a stage when the zippering fork in Grhl3 mutants begins to halt. Cellular protrusions are necessary to initiate cell-cell contact in late neurulation. 

However, ultimately, disparate transcriptional mechanisms are still likely to operate along the length of the neural tube axis, which will be commensurate with the phenotypic differences and resultant phenotypes seen in mice lacking Grhl3.

### 2.1. Brain—Neurogenesis, Morphogenesis and Cancer

The Grhl family plays a crucial role in the regulation of neural development across multiple species. There is a fine balance between stem cell survival and proliferation during the development of the CNS. In Drosophila, grh regulates the developmental program of neuroblasts by maintaining homeostatic levels of proliferation and apoptosis [56,57]. Neuroblasts go through multiple stages of proliferation during which they temporally cycle through different transcription factors. grh is switched on during the later stages of this process and continues to function during postembryonic regional patterning [56]. Interestingly, in grh mutants there are inconsistent changes in neuroblast proliferation depending on their location, indicative of a clear co-opting of disparate transcriptional mechanisms. Neuroblasts located in the thorax showed reduced proliferation, whereas higher proliferative rates were identified in the abdomen. This reinforces the central theme of our review, which serves to highlight that grh/Grhl-mediated transcriptional regulation is highly context- and tissue-specific. Notably, elegant time-course ChIP-SEQ analyses from defined timepoints of Drosophila embryogenesis suggest that GRH remains constitutively bound to the promoters of target genes, with subsequent transcriptional specificity conferred through the recruitment of transcriptional co-factors (both co-activators and co-repressors) [23]. However, the identification of these transcriptional partners currently remains largely unknown.

One of the best-characterised grh/Grhl targets is the transmembrane protein E-Cadherin (cdh1), a primary grh target essential for neuroblast proliferation [57]. E-cadherin is greatly reduced in grh-deficient post-embryonic neuroblasts and ectopic grh expression increases E-cadherin expression. Additionally, the E-cadherin promoter contains binding sites for Grh, providing the first evidence of a direct functional relationship between these two factors [57]. Subsequent studies have demonstrated numerous instances of grh/Grhl-cdh1 interaction (discussed later), indicating that this relationship is among the most important transcriptional pathways regulated by the grh/Grhl family.

Alongside expression in neuroblasts in the fruit fly, Grhl orthologues are also expressed in neural tissue during neurodevelopment in zebrafish (grhl2b) [13] and mice (Grhl3) [120]. In the zebrafish model, grhl2b is expressed within the midbrain-hindbrain boundary (MHB), and morpholino (MO)-mediated inhibition of grhl2b function led to apoptosis of neural cells, patterning defects in the MHB characterised by loss of MHB markers eng2a, pax2a, wnt1 and her5, and substantial disruption of MHB morphogenesis [13]. Chromatin-Immunoprecipitation (ChIP) assay demonstrated that eng2a was a direct target of grhl2b, and injection of eng2a mRNA rescued both the apoptosis and loss of MHB markers, but not the MHB morphogenesis, suggesting that MHB induction and morphogenesis were separate events, regulated by disparate genetic mechanisms [121]. 

To test that hypothesis, the study endeavoured to find a potential target gene that may be involved in grhl2b-mediated MHB morphogenesis. By using a customised dataset in which the promoters of the genes of all placental mammals, aligned to search for the presence of a conserved GRHL binding site (AACCGGTT), a novel target – small protein effector of cdc42, protein 1 (spec1) – was identified and confirmed as a direct relationship by ChIP. Spec1 was shown to interact with grhl2b to promote MHB morphogenesis, albeit not MHB patterning and induction, and rescued the MHB folding defects seen in the grhl2b morphants. Lastly, the study also explored potential regulatory mechanisms that operated upstream of grhl2b, and identified that the soluble morphogen fibroblast growth factor 8 (fgf8) worked together with grhl2b to regulate MHB patterning and morphogenesis upstream of eng2a and spec1. This is significant from an evolutionary perspective, as previous work in Drosophila had found that the fly fgf8 orthologue branchless could also activate grh, through phosphorylation of the serine/threonine protein kinase erk [54]. Whether the precise mechanism of activation is conserved in vertebrates remains to be determined, however this cross-species conservation of genetic pathways suggests a critically important regulatory pathway in the evolution of brain patterning and development.

Morpholino-mediated knock-down of grhl3 yielded similar MHB morphogenesis defects to those seen in the grhl2b morphants but did not lead to the same patterning and MHB induction defects [66], further confirming that the genetic mechanisms that drive patterning and morphogenesis are distinct. Consistent with these data, eng2a was not involved in the MHB folding defects seen in grhl3 mutants, however experimental data showed that both spec1 and arhgef19, known downstream genes, co-operate with grhl3 during MHB morphogenesis [13]. Collectively, these data within the brain indicate that grh/grhl factors are conserved between invertebrates and vertebrates to regulate important neurodevelopmental processes through different transcriptional targets. 

Further evidence of functional conservation of Grhl signalling in the brain came from detailed autoradiographic in-situ hybridisation (ISH) analyses of Grhl1-3 expression in the developing embryonic mouse brain [120,122]. Although Grhl1 and Grhl2 were never detected in neural tissue, Grhl3 was expressed in progenitor cells during early neurogenesis and restricted strongly and specifically to the habenula, a region that affects motor behaviour and decision making, in the final stages of embryogenesis [120]. Neural tissue-specific conditional deletion of Grhl3 (using Nestin^cre^-mediated recombination of a floxed-Grhl3 allele) resulted in aberrations in locomotor activity and decreased anxiety-like behaviours but did not show any defects in neurogenesis, neural patterning and neurotransmitter (tyrosine hydroxylase) release, stem cell proliferation or apoptosis [120]. Additionally, recent analysis of GRHL3 missense variants in an Iranian population suggests an underlying increase in risk of schizophrenia, although expression data and target gene analysis is unavailable. Nonetheless, these data are of interest for exploring the role of Grhl3 in the regulation of brain and behavioural function. Lastly, although Grhl2 is not expressed in neural tissue, one of the aforementioned Grhl2^−/−^ mice, Grhl2^clf3^, did display apoptosis of neural progenitors [16]. However, at present the mechanisms behind this apoptosis are unknown.

In brain cancer, particularly primary neuroblastoma, an early childhood neuroendocrine tumour originating from neural crest cells, GRHL1 functions as a tumour suppressor and its upregulation correlates with a favourable patient prognosis [39]. Genome-wide analysis of neuroblastoma cells identified 170 genes as potential GRHL1 targets; most of them are involved in nervous system development, cell adhesion, anchorage-independent growth, proliferation, differentiation, neuroblastoma genesis and spreading. Unfortunately, binding analysis of GRHL1 to the promoters of those genes was not reported in the study, precluding validation of their direct regulation [39]. With regard to GRHL2 and GRHL3, their role in brain cancer is yet to be explored.

### 2.2. Epidermal Development, Disease and Cancer

The mammalian skin is the largest organ in the body, forming a critical interface between the highly regulated internal milieu and the challenging environment. The outer-most layer of the skin epidermis, the stratum corneum, forms a front-line protective barrier that is established, maintained and repaired through the orchestrated regulation of structural and junctional proteins, lipids, enzymes and transcription factors [21,123]. Failure of the skin barrier to form through prematurity in humans or genetic manipulation in mice is associated with excessive fluid loss from within and susceptibility to microorganisms, toxins and allergens externally [124], and loss of grhl3 in zebrafish leads to defects in the formation of the outermost layer of the embryonic epidermis [66]. All three GRHL factors have been shown to contribute to skin development, barrier function and maintenance. They execute their functions through the regulation of unique and shared target genes with a specificity that is stage-dependent during organ development [125]. Mice deficient in Grhl1 display impaired cellular adhesion manifesting as defective hair anchoring and palmo-plantar keratoderma, resulting from the loss of Desmoglein 1 (Dsg1) expression, a direct GRHL1 target gene [70]. While there was a lack of genetic interactions between Grhl1 and Grhl3 in mouse embryonic skin development, skin wounds failed to heal in double heterozygous Grhl2/Grhl3 embryos and the loss of a single Grhl2 allele in Grhl3-null embryos results in their eyes being open at birth [125]. Both GRHL2 and GRHL3 directly regulate expression of the target gene ARHGEF19 in wound repair [75,125]. Homozygous mutations in GRHL2 are identified in patients with an autosomal-recessive ectodermal dysplasia syndrome and skin transcriptomic analysis demonstrated expression changes in genes implicated in cell-cell and cell-matrix adhesion networks [126]. The most striking skin phenotype is observed in Grhl3-null mice that fail to form a functional skin barrier during embryogenesis and die immediately after birth from excessive water loss [21]. This phenotype results from reduced expression of transglutaminase 1 (Tgm1), a bona fide GRHL3 direct target gene [20]. Postnatally, GRHL1 compensates for the loss of Grhl3 in skin barrier maintenance by inducing expression of Tgm5 in a safeguard mechanism that presumably evolved for terrestrial life [21], although this pathway has not yet been characterised in aquatic experimental species such as zebrafish or Xenopus.

The loss of Grhl1 in mouse skin deregulates desmosomal structures and results in a mild impairment of the skin barrier accompanied by an inflammatory response that predisposes the skin to squamous cell carcinoma (SCC) induced by chemical carcinogens (DMBA/TPA) [41]. While the Grhl3-null embryos are not viable, grafts of their embryonic skin onto immunocompromised NSG mice manifest phenotypic signs of inflammation, hyperproliferation and neoplastic transformation [127]. The GRHL3 factor also functions as a potent tumour suppressor against adult skin SCC. Conditional deletion of Grhl3 in mice (Grhl3cKO) using a keratin (K) 14-driven Cre transgene induces loss of its direct target gene Pten [28]. Interestingly, Grhl3cKO mice develop spontaneous papillomas and SCC, the onset and extent of which are markedly accelerated with exposure to DMBA/TPA [127]. In addition, promoting events such as the stimulation of an inflammatory response through administration of the sole tumour promoter TPA was sufficient to facilitate tumour development in Grhl3-deficient mice [26]. In primary human skin SCC, GRHL3 expression is repressed by microRNA-21 (miR-21) [26]. miR-21 also directly inhibits PTEN, and this synchronous regulation of PTEN and its direct transcriptional regulator GRHL3 results in oncogenic addiction to the PI3K/AKT/mTOR signalling pathway in cutaneous SCC [128,129]. 

### 2.3. Craniofacial Development and Head and Neck Cancers

An understanding of the role played by Grhl genes in craniofacial development can be traced back to the first report of grh in Drosophila [1], as the gene was originally named for the “grainy” and discontinuous chitinous head skeleton region seen in fly mutants lacking a functional grh gene. A number of direct grh targets were identified and subsequently characterised in these fly mutants, and these are listed in Table 1. Pioneering work from this model also identified the first known upstream regulators of Erk, those being the soluble morphogen branchless (orthologue of mammalian FGF8 [54]) and tyrosine kinase and extracellular signal-regulated kinase (erk) [44], highlighting the excellent genetic tractability and utility of this model for understanding conserved mechanisms of Grhl-dependent transcriptional regulation.

Since then, a role for GRHL-signalling in craniofacial development has been shown in multiple vertebrate models, and loss-of-function mutations are implicated in human epithelial disease that affects the head region, such as age-related hearing loss [130], ectodermal dysplasia [126], and both syndromic [14] and non-syndromic [131,132] clefts of the secondary palate, where Grhl3 is thought to interact with the known causative gene in Van der Woude Syndrome, Interferon Regulatory Factor 6 (IRF6). Importantly, craniofacial defects following loss of Grhl/grhl function have also been characterised in intermediary species between fly and human, including palatal clefting [17,133], premature suture apposition in the skull [134] and the lower-jaw defect micrognathia [18] in mice, while morpholino-mediated loss of grhl3 in zebrafish leads to defects in neural crest cell fidelity in the pharyngeal arches, with consequent hypoplasia of the lower cartilage seen in these morphants [7].

Although the craniofacial defects seen in mouse and fish models are pronounced, the genetic mechanisms by which these anomalies manifest have not been well defined. ChIP data and mRNA rescue experiments in zebrafish showed that the soluble molecule endothelin-1 (edn1) could substantially rescue grhl3-induced craniofacial hypoplasia in zebrafish, and co-injection of grhl3 and edn1 morpholinos (each at sub-phenotypic doses) led to additive phenotypes in the craniofacial cartilage [7]. Interestingly, genetic deletion of Grhl3 in mouse models has resulted in defects of both palate [14] and skull [134], albeit not the lower jaw, although somewhat intriguingly, tissue-specific loss of Grhl2 in the pharyngeal ectoderm and endoderm (via Sonic Hedgehog [shh]-Cre mediated deletion) led to comparable defects in lower jaw development [18]. Although no attempts have yet been made to rescue these mouse mandibular defects through Edn1 over-expression, genetic epistasis experiments conducted using mice double-heterozygous for Grhl2 and Edn1 (Grhl2^+/–^;Edn1^+/–^ mice) did not result in any mandibular defects, suggesting either that this Grhl-Edn1 mechanism is not conserved in mice, or alternatively, that 50% gene dosage remains sufficient for lower-jaw formation, and further reductions in this pathway may be necessary to observe deformities. In support of our second hypothesis is the fact that embryos lacking Edn1 (Edn1^−/−^ mice) also present with complete loss of the lower jaw [135], indicating that both Grhl2 and Edn1 are critical genes in driving mandibular patterning and development.

Regarding formation of the secondary palate in the upper jaw, again Grhl2 appears to be the predominant orthologue regulating fusion of the palatal shelves. Numerous Grhl2-deletion mouse models show significant clefting of the face, abdominal cavity, cranial neural tube and posterior spinal cord [14,15,16,105,107], although in most cases these embryos die by mid-gestation. However, Grhl2^−/−^ embryos that survive to E18.5 invariably present with cleft palate [16,107], suggesting that Grhl2 is a conserved factor in the regulation of tissue fusion across numerous organs and epithelia across embryonic development. 

Supporting the involvement of Grhl2 function in palatogenesis was the recent discovery of the first known craniofacial-specific genomic enhancer element (termed mm1286) that was found to not only drive Grhl2 expression specifically within the craniofacial primordia, but importantly, also led to cleft palate when deleted in a mouse model expressing only one allele of Grhl2 (i.e., Grhl2^+/−^;mm1286^+/−^ mice) [17]. Interestingly, this 2.4kb enhancer region comprises a highly conserved 325bp consensus sequence that is both necessary and sufficient for mediating the craniofacial-specific enhancement of Grhl2 transcription, and within this region lies a 12bp sequence that is invariant amongst all vertebrate species examined. This 12bp region contains putative binding sites for a number of homeobox genes—Prep1, Meis1, Meis2, MRG2 and TGIF2—and introducing mutations into this 12bp region substantially reduced the craniofacial specific activity of the mm1286 enhancer at E11.5 [17]. These data suggest that homeobox-factor mediated transactivation of Grhl2 may be a key novel pathway in Grhl2-dependent regulation of palatal development.

Although Grhl2 in the palatal shelves is expressed exclusively in the epithelium (and potentially periderm) surrounding the developing palatal shelves [17,122], nonetheless evidence exists that Grhl2 can indirectly regulate genes in the underlying palatal mesenchyme as well. An example of this is the transcription factor Zeb1, whereby inactivation of ZEB1 signalling can significantly ameliorate both palatal clefting and embryonic lethality in Grhl2^−/−^ mice [133]. Consistent with known roles of Grhl2 maintaining epithelial identity in cancer models, Grhl2^−/−^ embryos present with premature EMT within the palatal epithelium of the developing shelves, and this appears to either underpin, or be correlated with, impaired palatal migration and fusion. The intermediate mechanisms that are involved in mediating the Grhl2-epithelial signal to the mesenchyme remain to be elucidated, although expression data from control and Grhl2^−/−^ epithelia suggest that the genes Ovol1, Ovol2 and the micro-RNA miR-200 family are critical to maintaining epithelial fidelity within the palate. 

Skin and head and neck stratified epithelia shared developmental similarities that prompted analysis of the GRHL factors in these tissues. GRHL3 was shown to regulate stage-specific development of the circumvallate papilla of the mouse tongue during organogenesis, not just by maintaining epithelial integrity but also by regulating epithelial cell proliferation, apoptosis, and migration [136]. The effect of Grhl3 knockdown is likely to be mediated by altered expression of numerous signaling and apoptosis-related molecules, which include Axin2, Bak1 (BCL2-antagonist/killer 1), Bcl2 (B cell leukemia/lymphoma 2), Casp3 and 8 (caspase 3 and 8), Ccnd1 (cyclin D1), Ctnnb1 (catenin [cadherin associated protein], beta 1), Gli3 (GLI-Krüppel family member GLI3), Lef1 (lymphoid enhancer binding factor 1), Ptch1 (patched 1), Rock1 (Rho-associated coiled-coil containing protein kinase 1), Shh (sonic hedgehog) and Wnt11 [136]. In adult mice, GRHL3 maintains the homeostasis of the oral epithelium by regulating the balance between proliferation, differentiation and terminal differentiation [137]. Interestingly, loss of Grhl3 induces spontaneous head and neck cancer in aged mice through loss of its direct target gene, GSK3b [27]. Moreover, GRHL3 expression in a subset of head and neck cancer patients induces tumour differentiation through induction of its tumour-specific target gene FLG and this differentiation potentially influences anti-cancer therapy responses and patient prognosis [137]. Contrarily, GRHL2 induces an epithelial phenotype and acts as an oncogene by regulating stemness in oral cancer cells [95]. The knockout of Grhl2 in mice abolished carcinogen (4-NQO)-induced oral tumours and this protective mechanism was explained by the loss of GRHL2 target genes E-cadherin, hTERT, p63, and miR-200 [138]. The role of GRHL1 and its target genes in head and neck development and cancer is still unknown.

### 2.4. Kidney—Development and Cancer

The functions of GRHL factors in kidney development and disease have been summarized in a recent review [139]. Briefly, all three Grhl1-3 genes are expressed in the developing kidney, in the proximal tubules (Grhl1), distal tubules (Grhl2), collecting ducts (Grhl1 and Grhl2), nephric ducts (Grhl2), renal pelvis (Grhl1 and Grhl3), ureter (Grhl3) and urothelium (Grhl2 and Grhl3) [139]. Grhl1-null mice do not display any direct kidney phenotype, but they are prone to the development of salt-sensitive hypertension, which suggests renal malfunction [140]. The expression of Cacna1d (calcium channel, voltage-dependent, L type, alpha 1D subunit), Ddc (dopa decarboxylase), Flot2 (Flotillin 2) and Xpo1 (Exportin 1) is reduced in the Grhl1-null kidneys, and all these genes have potential GRHL-binding sited in their promoter regions [80]. Cacna1d, Ddc, Flot2 and Xpo1 have previously been linked to the regulation of blood pressure, which may explain the salt-sensitive hypertension in Grhl1-null mice [80]. Renal development is severely impaired in Grhl2-deficient mice, with defective lumen expansion and barrier defects in embryonic kidneys [73] as well as defects in epithelial barrier, urine concentration and medullary osmolarity in adult mice [74]. The relevant GRHL2 target genes may include Cdh1 (cadherin 1), Cldn4 (claudin 4), Myh10 (myosin, heavy polypeptide 10, non-muscle), Ovol2 (ovo like zinc finger 2), Rab25, Shroom3 and Tjp3 (tight junction protein 3) [73,74].

However, the above-mentioned review [139] did not address the involvement of these factors in cancer. Historically, the first such report concerned the potential application of GRHL2 as a new prognostic biomarker in clear cell renal cell carcinoma (ccRCC) [141]. This study revealed rather complex function of GRHL2 in ccRCC. On one hand, GRHL2 mRNA expression was reduced in tumor samples, and lower GRHL2 levels were associated with significantly better survival, indicative of its tumor suppressor role. On the other hand, GRHL2 protein expression was positively associated with higher tumor grade, larger tumor size and higher risk of disease relapse. Yet, GRHL2-positive patients had significantly lower disease-free survival than those who were GRHL2 negative [141]. Our own studies revealed that both GRHL1 and GRHL2 serve protective roles in the development of ccRCC [142]. Their expression is reduced in ccRCC samples and is correlated with the expression of von Hippel-Lindau (VHL) gene, a key tumor suppressor of ccRCC [143]. When the expression of GRHL2 was silenced in RC-124 non-tumorigenic kidney cell line, this increased cell proliferation and reduced apoptosis, further supporting the tumor suppressive role of GRHL2. In the same study, we found no indication that GRHL3 may be involved in ccRCC [143].

### 2.5. Breast—Development and Cancer

Within normal adult breast tissue, GRHL2 is localised to the luminal epithelia and basal myoepithelial cells of the ductal-lobular unit [144]. Grhl2 is expressed in the luminal epithelial and myoepithelial cells during the early stages of ductal development in mice [145]. This study went on to look at potential target genes of Grhl2 during breast development. Using linear regression to compare mRNA expression, the authors concluded that Grhl2 expression in the breast correlated with expression levels of E-cadherin, Claudin 3 (Cldn3), 4 (Cldn4), and 7 (Cldn7) which are all important to establish functional mammary ducts. Although E-cadherin and Cldn4 are confirmed GRHL2 target genes in several types of murine epithelia [15], it is important to keep in mind that GRHL2-gene interactions are dependent on timepoint, tissue type and context so further studies are needed to confirm that these genes are direct GRHL2 target genes. Although there have been no other studies characterising Grhl2 or any other Grhl family members during breast development, one can begin to draw some parallels from existing breast cancer studies that detail a role for GRHL2. 

Erbb3 shares a similar expression pattern to Grhl2 during breast cancer in the luminal epithelium. shRNA mediated knock-down of Grhl2 in breast cancer cells caused the down-regulation of Erbb3 resulting in lowered levels of cell proliferation and changes associated with EMT [34]. In the context of breast cancer, GRHL2 induces an epithelial phenotype in mammary cells and its expression prevents tumour initiation in breast cancer xenografts, sensitises breast cancer cells to chemotherapy, and suppresses the emergence of treatment-resistant cancer stem cells [146]. GRHL2 regulates the estrogen receptor signalling in breast cancer by directly regulating the expression of ER target genes [147]. This transcription factor was also shown to induce expression of the membrane receptor LYPD3, and the LYPD3 ligand, AGR2 to promote the growth of endocrine therapy-resistant breast cancers in mice [148]. Furthermore, the knockdown of GRHL2 in breast cancer cells led to the induction of the TGF-β/Smad target gene, CTGF [10]. Here, GRHL2 suppressed EMT mediated by the TGF-β signaling pathway and it was downregulated in EMT-dependent mammary tumours (claudin low, metaplastic) and cell lines (basal-B). The mechanism by which GRHL2 regulates EMT is largely thought to occur through repression of its direct target gene, ZEB1 [10]. Interestingly, ZEB1 also directly regulates GRHL2 expression in a reciprocal feedback loop that forms an interplay between EMT and MET. The GRHL3 factor is believed to suppress breast cancer growth [28] and to induce basal mammary cell epithelial differentiation [149]. However, the GRHL3 direct target genes involved in these processes are still unknown. The role of GRHL1 and its target genes in breast development and cancer is still unknown.

## 3. Lung—Development and Cancer

Grhl2 is strongly expressed in early development of the murine lung. By E16.5 Grhl2 is expressed in the tracheal epithelium and throughout epithelial cells lining both large and small bronchioles [122]. After this timepoint, expression levels of Grhl2 dramatically decrease. In the human lung GRHL2 is expressed in p53+ and KRT5+ basal airway epithelium progenitor cells. In comparison, GRHL1 is expressed in MUC5AC+ goblet cells whereas GRHL3 expression is found in tubulin+ ciliated cells [11]. The epithelium lining the airways within human lungs is composed of undifferentiated basal cells that maintain ciliated secretory luminal cell populations. Several hundred likely GRHL2 gene targets were identified in primary human bronchial epithelial (HBE) cells. RNA-Seq compared gene expression of control undifferentiated HBE cells from three human donors compared to HBE cells with a mutated GRHL2 protein (created by inducible lentivirus protein). Gene ontology analysis revealed that absence of GRHL2 in HBE cells was responsible for significant down-regulation of cell adhesion, morphogenesis and migration genes. To elucidate which genes were directly downstream of GRHL2, ChIP seq was performed on comparable HBE cell samples from the three human donors. Known targets CLDN4 and CDH1 in addition to MPZL2, KRT5, SEMA4B and HMBB were identified as downstream targets of GRHL2. These cells were then grown in air-liquid interface (ALI) culture allowing them to differentiate and to assess barrier function. In cells containing the mutated GRHL2 protein, differentiation was aberrant and barrier formation failed to be established. Further RNA-seq analysis confirmed down-regulation of a number of genes including RHBDL2 (up-regulated in epidermal cells during wound repair), SCRIB (crucial for planar cell polarity), RAP1A (involved in cell locomotion), MZVL2 (a Grhl2 target in murine ectoderm), PVRL4 (important for the adherens junction formation in skin), CGN, CGNL1 and CTNND1 (cell adhesion genes). Together the findings from this study showed that GRHL transcription factors and their target genes play an integral role in differentiation from basal progenitors, cell adhesion and barrier formation in human mucociliary epithelium [11]. 

Building on these findings, CRISPR/Cas9 genome editing was used to further address the function of Grhl2 in the developing lung. Consistent with previous findings, lack of Grhl2 in the majority of epithelial cells within a tissue or in culture was detrimental to columnar morphology and barrier permeability. Notably, this was not seen in regenerated mice trachea following conditional knock-out of Grhl2 in ~30% of basal progenitors, suggesting that these unaffected cells can compensate in vivo. Concomitantly, expression of Notch pathway components was also investigated in organoid cultures, where mutant spheres showed significantly higher expression of Notch1, Jag1 and Jag2, and significant down-regulation of Notch3 and Dll3, suggesting that Grhl2 directly mediates Notch pathway signalling in airway epithelium. CRISPR/Cas9 genome editing was then used to generate mutations in previously identified Grhl2 target genes in two different assay cultures [11]; the ALI culture (as described in [11]), as well as clonal organoid (bronchosphere) culture. Colony forming efficiency in cells mutant for ANKRD22, PVRL4, SMAGP and ZNF750 was significantly lower. The study then went on to focus on the two novel Grhl2 targets identified, namely Smagp and Znf750 to elucidate their function in airway mucociliary epithelium. Mutation of SMAGP disrupts barrier formation and organoid cultures showed disturbed luminal morphology within the bronchospheres. Clonal expansion was not disrupted when Grhl2 was conditionally deleted in basal progenitors but the cells’ ability to differentiate into multiciliated luminal cells was disturbed. Consistent with this, mutations in ZNF750 inhibited ciliogenesis and similarly to SMAGP also delayed barrier formation, consistent with Znf750 being a direct downstream target of Grhl2. 

Recent findings have also been reported in a novel mouse line where lung epithelial driver Shh-Cre was used to conditionally delete Grhl2 [19]. At E18.5 Grhl2^Δ/−^ embryos exhibited thickened hypercellular intersaccular septa, reduced distal saccule size and a larger condensed density of Nkx2.1+ cells likely due to the smaller size of the lungs compared to controls. Consistent with previous findings, absence of Grhl2 impaired the ability of basal progenitor cells to differentiate into ciliary luminal cells. Airway precursor cells were not affected by lack of Grhl2, however numbers of Sox-9+ distal progenitor cells were still detected in the conditional Grhl2 mutants compared to the scarce numbers found in control embryos [21]. The delay in epithelial differentiation is the likely mechanism behind the markedly higher number of Sox9 expressing cells found in Grhl2^Δ/−^ embryos. RNA-seq was used to identify Grhl2 targets that may be responsible for this delay in epithelial differentiation. Gene ontology analysis showed that cilia development genes were significantly downregulated in the mutants, reinforcing the importance of Grhl2 in ciliogenesis. Surprisingly no changes in Notch signalling, known for its role in cilia development [89], could account for this phenotype. However, Multicilin and Znf750 (previously described in [89]) were both significantly downregulated in the Grhl2^Δ/−^ mutants. A novel target gene also found in this study is Elf5. mRNA and protein expression analysis confirmed that Elf5 is significantly reduced in the Grhl2^Δ/−^ embryos and ChIP revealed specific binding of Grhl2 to the Elf5 promoter. Previous studies have found that overexpressed Elf5 promoted dilated epithelia and disturbed branching morphogenesis during early lung development [150]. Collectively, these studies show both overlap in previously identified Grhl target genes as well as novel targets. 

In the context of lung cancer, a recent study identified upregulation of GRHL1 in non-small cell lung cancer (NSCLC) that correlates with poor patient survival. The authors showed that GRHL1 binds to the promoters of cell cycle-related genes CDC27, RAD21, CDC7, and ANAPC13 to induce their transcription, while GRHL1 knockdown inhibits NSCLC tumour growth [100]. 

Loss of GRHL2 in airway epithelial cells results in the downregulation of its direct target gene ESRP1, a central coordinator of alternative splicing processes in the regulation of EMT. Reduced ESRP1 downstream of GRHL2 leads to the induction of p120-catenin and loss of E-cadherin [138]. In lung cancer cells, increased GRHL2 expression promotes cell growth and colony formation, and simultaneously suppresses cell migration by directly regulating its target gene, RhoG [96]. Although GRHL3 is known to function as a tumour suppressor of SCC [151], it remains to be seen whether this role is conserved in lung SCC and other lung cancer subtypes.

## 4. Liver—Development and Cancer

Grhl2 is expressed in cholangiocytes (epithelial cells lining the bile ducts) which originate from fetal liver stem cells. Cholangiocytes are responsible for regulating the composition and volume of the bile travelling through the hepatic bile ducts [152]. The secretory and barrier functions of cholangiocytes are essential for the functional development of bile ducts. Within the biliary ducts, tight junctions are crucial to avoid any leakage of bile as it travels through the liver. Tight junctions begin to form shortly after the differentiation of hepatoblasts into cholangiocytes. Gene expression profiles from 3D cultures showed that Grhl2 was specifically expressed in cholangiocytes and not hepatoblasts derived from liver progenitor cells [87]. These 3D cultures were generated using a liver progenitor cell line (HPPL), to form cyst-like structures that have a lumen surrounded by polarized epithelial cells. Interestingly, when compared to the cysts formed from adult cholangiocytes, the cysts from the HPPL cells had significantly smaller lumens. When over-expressed, Grhl2 enabled the HPPL cells to form larger lumens. Co-localisation analysis by targeting Cldn3 and Cldn4 expression, suggested that GRHL2 was able to drive epithelial morphogenesis resulting in the enlargement of the apical lumen. However, the expression of Cldn3 and Cldn4 alone was not enough to increase lumen size in line with Grhl2 overexpression. To identify an additional factor necessary for this process, both human and mouse databases were searched for genes that have a Grhl2 binding sequence (AACCGGTT) [17] and these were then cross-referenced with the micro-array data used previously. Using this approach, Rab25 was identified as a possible target and found to localise Cldn4 to the tight junction. HPPL-Rab25+Cldn3 3D cultures indeed had larger lumens than HPPL-Cldn3+Cldn4, however it was still not equal to the size of HPPL-Grhl2. These findings suggest more research is required to elucidate a possible additional factor regulated by Grhl2 important for the function of tight junctions in this system [87]. 

We previously described the importance of Grhl2 for the differentiation capacity of distal progenitor cells in the lung. Conversely, in the liver we see that differentiation is inhibited in the presence of Grhl2 [79]. Together this shows the spatial heterogeneity of Grhl2 role in different tissue types. Comparison of neonatal and adult cholangiocytes, showed that Grhl2 which was downregulated in neonatal cells but expressed in adult cells was inhibiting the differentiation of adult cholangiocytes. By supplementing Grhl2 whilst inducing hepatocyte differentiation, hepatocyte markers including albumin, CPSI, G6Pase were inhibited in neonatal cells [81]. Furthermore, the downregulation of Grhl2 by shRNA in adult cells increased their hepatocyte characteristics. miRNA analysis has since revealed that miR122 was significantly downregulated in hepatic progenitor (HPPL) cells in vitro [90]. Specifically, GRHL2 negatively regulates the promoter of miR122 and is thought to be a downstream effector target disturbing the differentiation potential in the presence of Grhl2. Consistent with other tissues we have discussed, this does not appear to be the only target gene of Grhl2 in this scenario. Another possibility is that Grhl2 is influencing the Notch pathway. In accordance with this hypothesis, upregulation of Grhl2 and Notch, respectively, have both shown to inhibit hepatic differentiation [153]. In hepatocytes, Notch signalling has been shown to regulate transcription factors including HNF1β, HNF1α and HNF4 each important for hepatocyte differentiation [153]. Further research is required to elucidate whether Grhl2 targets components of the Notch signalling pathway during hepatocyte differentiation. This role of Grhl2 in the developing liver highlights two important points; firstly that Grhl2 is important for hepatocytic differentiation. Secondly, this reiterates the highly diverse functions of Grhl factors and how their function is dependent on spatial and temporal expression.

Relatively little is known about the Grhl factors in liver cancer. Of the GRHL1-3 factors, GRHL2 acts as an oncogene in hepatocellular carcinoma (HCC) and HCC cells show decreased proliferation following GRHL2 downregulation. In addition, the gain of GRHL2 on 8q22.3 in HCC patients is associated with intrahepatic metastasis and early recurrence [31]. GRHL2 levels are increased in patients with alcoholic liver disease in which this factor suppresses miR-122-dependent differentiation and metabolism in mice and humans [154]. Moreover, SNPs within the GRHL2 locus are identified in patients with chronic hepatitis B and HCC [155]. The role of GRHL1 and GRHL3 in liver cancer remains unknown.

### 4.1. Gastrointestinal Tract and Bladder—Development and Cancer

It seems likely that all three GRHL1-3 factors play important roles in the development of the gastrointestinal tract. In mouse embryos, Grhl1 and Grhl3 are highly expressed in the esophagus and fundus of the stomach, as well as in the anal canal. Grhl2 is weakly expressed in the upper gastrointestinal tract and in the midgut, and both Grhl2 and Grhl3 are detectable in the distal hindgut [122]. Despite the above observations, very few studies were dedicated to this topic. In the murine bladder epithelium, GRHL3 is a downstream target of Krüppel-like 5 (KLF5) and is required for the maturation of this epithelium [156]. Grhl3-null mouse embryos display defects in the terminally differentiated umbrella cells, and these abnormalities are likely to be mediated by the altered expression of putative GRHL3 target genes, such as Cldn5, 6, 8, 9 and 11; Ocln (occludin) and Upk2 (uroplakin 2) [72]. Of these, at least Upk2 is directly regulated by GRHL3 [72].

The results of studies carried out in both mouse models and human patients indicate that GRHL3 acts as a tumor suppressor in head and neck squamous cell carcinoma (HNSCC) [27]. Its levels are reduced in human HNSCC samples compared with adjacent normal tissues. Conditional deletion of Grhl3 in oral epithelium in mice resulted in the loss of Gsk3b (glycogen synthase kinase 3 beta) expression and predisposed them to chemically induced HNSCC [27]. GRHL2 performs a tumor suppressor role in gastric cancer [94,157]. Its expression is significantly downregulated in gastric cancer samples obtained from human patients. Ectopic expression of GRHL2 in a gastric cancer cell lines inhibited proliferation, invasion and migration, and promoted apoptosis. Relevant GRHL2 target genes may include Bcl2, c-Myc and Mmp2, 7 and 9 (matrix metalloproteinases 2, 7 and 9) [94,157].

Tumor-promoting roles of GRHL2 in oral squamous cell carcinoma (OSCC), esophageal, pancreatic and colorectal cancer (CRC), as well as similar roles of GRHL3 in CRC, have been summarized in a recent review [25]. GRHL2 target genes relevant for OSCC development include FoxM1B (forkhead box M1B), miR-200 family, Oct4 and TERT (telomerase reverse transcriptase) [85,93,138], while in CRC ZEB1 (zinc finger E-box binding homeobox 1) is such a target [34]. In a recent study, all three GRHL1-3 transcription factors were shown to perform oncogenic functions in CRC [158]. The expression levels of all three GRHL1-3 genes were found to be higher in CRC samples than in normal colon tissues. Low levels of GRHL1-3 expression conferred a better overall survival of patients while high levels of their expression were associated with poor disease-free survival. Knockdown of GRHL1 or GRHL2 or GRHL3 significantly reduced the proliferation and inhibited the ability of colony formation of human CRC cells [158].

All three GRHL1-3 genes are frequently mutated in human bladder cancer [25]. GRHL2 acts as a tumor suppressor in this type of cancer, where it inhibits EMT by targeting ZEB1 [159]. Moreover, expression of GRHL2 is decreased in human bladder cancer samples, and downregulation of GRHL2 increases the proliferation rate of bladder cancer cells [159]. GRHL3 also plays a role in bladder cancer. Its expression is decreased in undifferentiated bladder cancer cells. GRHL3 impairs migration and invasion of bladder cancer cells, but does not influence their proliferation rate [160].

### 4.2. Development and Disease in Non-Epithelial Tissues

Although predominantly thought of as being an “epithelial”-specific transcription factor family, nonetheless several lines of evidence indicate that this is not always the case. In addition to the aforementioned expression in the developing brain of Drosophila [57], zebrafish [13] and mice [120], Grainyhead-like gene expression in higher-order vertebrates has been described in several non-epithelial tissues, and these will be summarised briefly below.

## 5. Endothelium

Within the endothelium lining blood vessels, Grhl3 is expressed and regulates endothelial cell growth, proliferation and apoptosis [86,99], partially through the regulation of Akt and endothelial nitric oxide synthase transcription. The specificity of this pathway is highlighted by the fact that a common regulatory factor in endothelial tissue—VEGF—does not influence Grhl3-mediated activity in these cells, although Grhl3 function is influenced by supplementation of cultured cells with exogenous nitric oxide [86]. 

Moreover, several deletion mouse models of the family member Grhl2 indicate apparent vasculature or endothelial defects in the heart [107] and lungs [18], which suggests that that these may also be underpinned through defective embryonic development of the endothelium.

### Placenta

GRHL2 is highly expressed in the developing extra-embryonic placental tissue, particularly in mouse basal chorionic trophoblasts. Although somewhat epithelial in morphology, and critical for embryonic growth, these cells are nonetheless not derived from a germinal layer during development, and do not contribute to final embryonic cellularity. Nonetheless, studies have shown that loss of Grhl2 and subsequent downregulation of the direct target gene Spint1 led to disruptions in polarity, morphogenesis, basement membrane integrity and branching of the developing maternal-fetal interface [81]. Moreover, this study also identified a number of other target genes to which Grhl2 is bound (as demonstrated by ChIP assay), indicating that further transcriptional networks also contribute to the branching morphogenesis within this tissue [81]. As an aside, however, Grhl2-mediated regulation of Spint1 was also identified as a novel mechanism driving salivary gland development [82], a tissue also characterised by extensive branching, suggesting that the Grhl2-Spint1 pathway may be conserved in the establishment of morphogenetic branching complexity across multiple organ and tissue types.

## 6. Bone (Osteoblasts)

Grhl3 expression has previously been described in the developing limb mesenchyme [67], and also within developing endochondral and subchondral in the murine limbs [161]. Recently, Grhl3 upregulation was shown to be a key player in the transcriptional program of differentiation in early osteoblasts, where it was up-regulated >300-fold in osteoblast progenitors treated with a combination of BMP2 and wnt3a [161]. Promoter analysis indicated two consensus Tcf/Lef binding sites, suggesting a clear regulatory link between canonical Wnt signalling and Grhl3 induction. Moreover, a direct linear relationship was apparent between levels of BMP2 protein and Grhl3 transcription in vitro, suggesting that BMP2 is also a direct upstream regulator of Grhl3. Lastly, mouse fracture healing studies also showed a strong upregulation of Grhl3 in bone repair [161], consistent with known roles in repair of the epidermis [75]. Taken together, these data clearly show a BMP/Wnt-dependent role for Grhl3 in differentiation and repair of a non-epithelial tissue.

It is clear from mounting experimental data that the Grainyhead-like genes are among the most critical transcription factors that govern both embryonic development and post-natal diseases, particularly cancer. Although primarily exerting regulatory roles in epithelial tissue (and by extension, de-regulation contributes to adult cancers of epithelial origin), it is clear that the roles of these genes are wider than first thought, with transcriptional regulation evident in numerous non-epithelia tissues as well. Intriguingly, much data suggest that there are several genes that are regulated by the Grhl family across different embryological and adult contexts (e.g., Cdh1, Cldn4, Elf5 and Zeb1); overwhelmingly large-scale transcriptomic data suggest that Grhl-Target interactions are specific, and context-based. Future studies should aim to characterise the involvement of targets implicated in one process (e.g., edn1 in craniofacial development) across multiple other processes, to determine which of these pathways are core regulatory mechanisms involved in the most basic steps of tissue organisation and maintenance. Through the ongoing evolution of “big-data” (single cell RNA-SEQ, spatial transcriptomics [162], ChIP-SEQ, bioinformatic meta-analyses [42] etc.) combined with an increased focus on identifying both upstream regulators and transcriptional co-factors of the Grhl family, the future of understanding the Gene Regulatory Networks (both critically conserved and tissue/organ/disease specific) is likely to be substantially improved in the coming years.

## Figures and Tables

**Figure 1 ijms-23-02735-f001:**
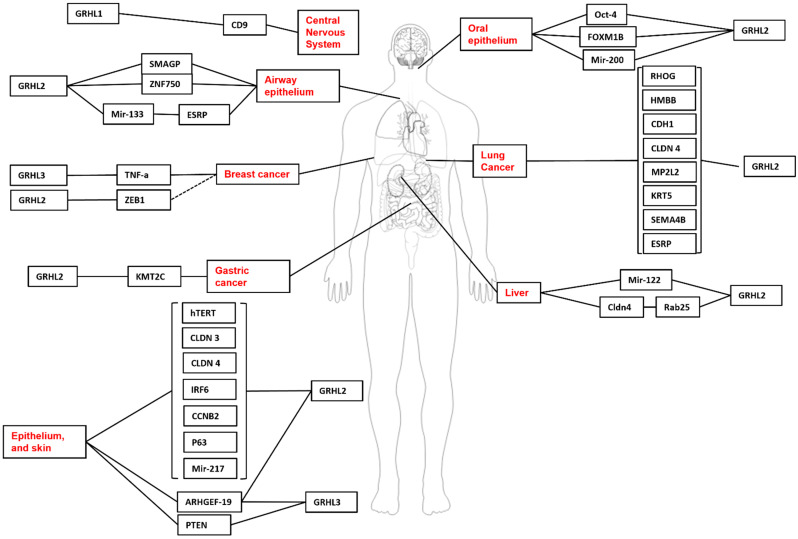
Grainyhead-like target genes in human development and disease. GRHL1 upregulates CD9 in the development of neuroblastomas. In Airway epithelium, GRHL2 regulates the expression of SMAGP and ZNF750 in airway cell polarity, barrier maintenance and cell differentiation. Similarly, mir-133 is regulated by GRHL2 and controls EMT in the airway epithelium through its own regulation of ESRP. A host of genes are regulated by GRHL2 in lung tumourigenesis, including HMBB, CLDN4, CDH1, MP2L2, KRT5, SEMA4B, ESRP and RHOG. GRHL3 regulates TNF- in endothelial cell migration in breast cancer cell lines and PTEN in squamous cell carcinoma. GRHL2 is also involved in cancer progression, regulating KMT2C in gastric cancer lines. Further, GRHL2 also regulates Oct-4, Mir-200 and Foxm1b in oral epithelium, as well as mir-122 in the differentiation of hepatocytes and RAB25 in liver cancer. GRHL2 regulates various genes with processes in the epithelium, including hTERT, Claudin 3 and 4, IRF6, CCNB2, p63, Mir-218. Both GRHL2 and GRHL3 co-operate in the regulation of ARHGEF19 in epidermal wound repair.

**Figure 2 ijms-23-02735-f002:**
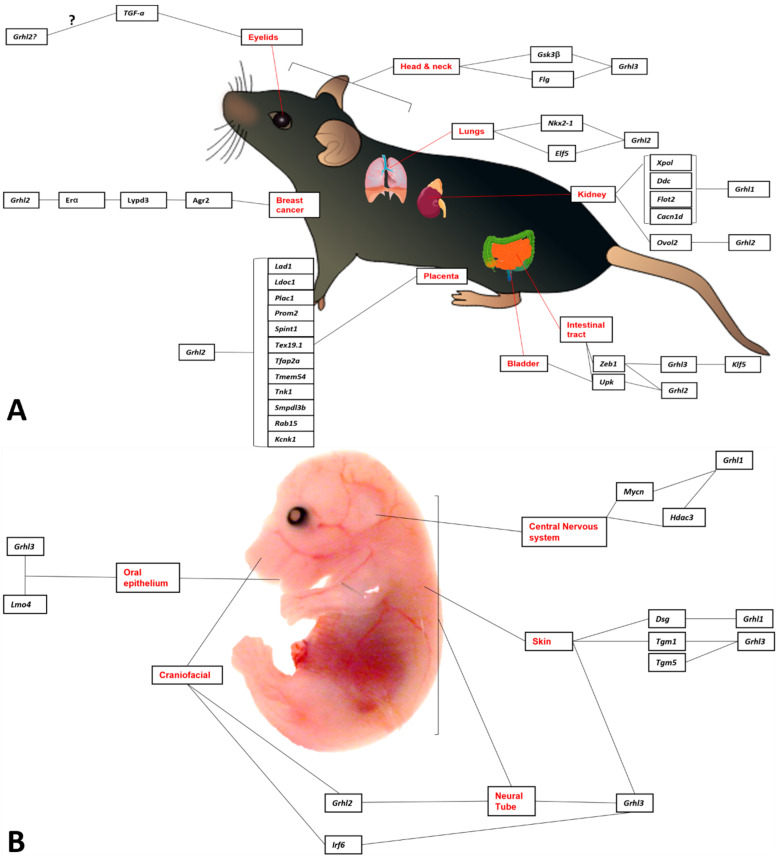
Grainyhead-like target genes in embryonic and adult murine tissue. (**A**) In adult mice and mouse-derived cell lines Grhl2 binds to Elf5 in embryonic lung tissue and Nkx2-1 in the lung epithelium specifically. In the kidney, Ovol2 is positively regulated by Grhl2, whilst in the Bladder Grhl2 regulates Upk in the differentiation of umbrella cells aiding in barrier formation. Grhl2 also regulates TGF-a in eyelid development. A number of genes have been found to have been regulated by Grhl2 in the placenta, including Lad1, Ldoc1, Plac1, Prom2, Spint1, Tex19.1 Tfap2a, Tmem54, Tnk1, Smpdl3b, Rab15 and Kcnk1, although the functions of these interactions remain unclear. In the kidney, Grhl1 regulated cacna1d, Ddc, Flot2 and Xpol in tumourigenesis. Similarly, Grhl2 regulates a number of genes in breast cancer including Erα and ZEB1 in the context of epithelial-mesenchymal transition. Grhl3 regulates Upk in the bladder and ZEB1 in the gastro-intestinal tract, whilst itself regulated by Klfs in the gut. (**B**) Mouse embryo target genes. Grhl1 regulates Dsg in hair development and Mycn and Hdac2 in the promotion of tumourigenic properties in neuroblastoma lines. Grhl3 controls TGM1/5 in skin barrier defects and Lmo4 in the oral epithelium.

**Figure 3 ijms-23-02735-f003:**
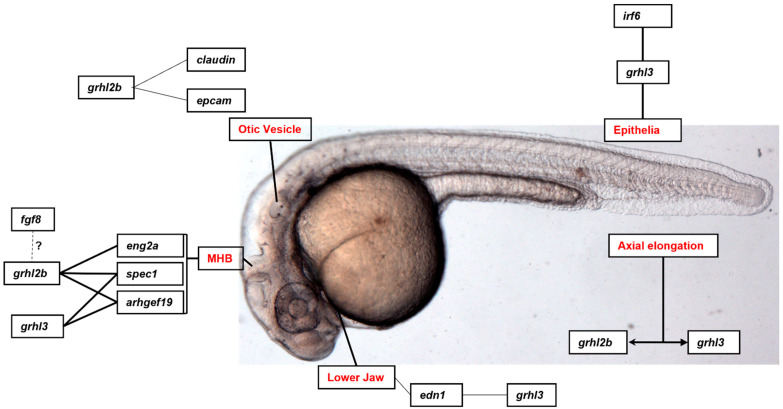
Grainyhead-like target genes in zebrafish development. grhl3 regulates edn1 in the development of the lower jaw in zebrafish embryos, and interacts with grhl2b in axial extension of the zebrafish tail. Irf6 governs grhl3 expression which in turn regulates keratinocyte differentiation. grhl2b regulates the development of the otic vesicles via claudin and epcam, as well as the MHB and neuroblast development and cancer via eng2a, spec1 and arhgef19. Additionally, fgf8 is a putative regulator of grhl2b regulation of the MHB.

**Figure 4 ijms-23-02735-f004:**
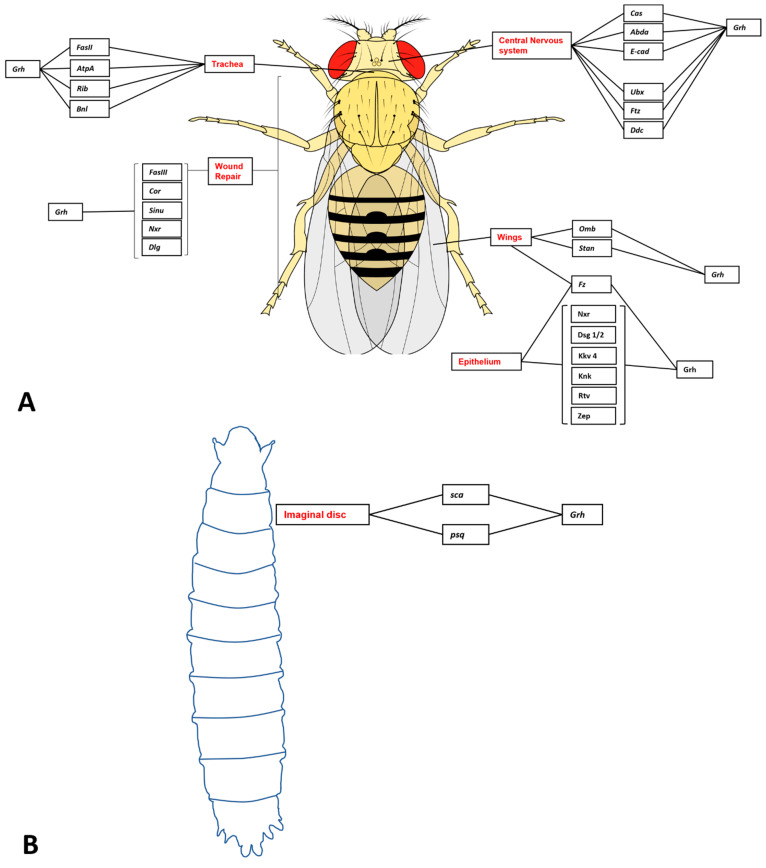
Grainyhead-like target genes in Drosophila adults and larvae. (**A**) Adult Drosophila target genes. Grh regulates tracheal branching through the target genes FasII, AtpA, Rib and Bnl. Expression of Grh in the amnioserosa results in tissue closure defects as a result of regulation of FasII, Cor, Sinu, Nxr and Dlg. Grh also regulates Cas in the differentiation of neuroblasts, as well as Abda and E-cad. Grh also regulates Ubx, Ftz and Ddc by binding to neurogenic enhancers on these genes. Grh regulates Stan and Omb in wing development, and the absence of these genes results in a ‘notched’ wing phenotype. Grh regulates a number of processes in the epithelium, including wound repair, junction and barrier integrity and development. This is achieved through the genes Nxr, Dsg1/2, Kkv, Knk, Rtv and Zep. (**B**) Grh target genes in Drosophila larvae development. Grh regulates the genes Sca and Psq, which are responsible for development of the imaginal disc and by extension, the development of the adult fly from the larvae following metamorphosis.

**Table 1 ijms-23-02735-t001:** Known or hypothesised direct target genes of *grh* characterised in invertebrate species, namely the fruit fly Drosophila, the honeybee (*Apis mellifera*) and the nematode model C. elegans.

GENE(S)	SPECIES	MAMMALIAN	BIOLOGICAL ROLE	REFERENCE(S)
		ORTHOLOGUE		
*Ultrabithorax (Ubx)*	Drosophila	*HoxB6*	*Grh* can bind to and/or directly activate transcription from the promoters of these genes during the development of neuronal cells.	[44,45]
*Dopa Decarboxylase (Ddc)*		*DDC*		
*Fushi-tarazu* *(Ftz)*	Drosophila	*NR5A2*	*Grh* regulates *Ftz* during epidermal wound repair.	[44,46]
*Suppressor of Hairless (Su (H))*	Drosophila	*CBF1*	Modulation of *Notch* signalling in regulation of cell fate.	[47]
*Torso (tor)*	Drosophila	*TEK receptor tyrosine kinase*	The *torsos* receptor tyrosine kinase (RTK) can remove *Grh*-mediated repression of the tailless gene during development.	[48]
*Tailless (tll)*		*NR2E1*		
*Zerknullt (zen)*	Drosophila	*Hox A3/B3/D3*	*Grh* binds to a *zen* repression element in a ventral repression region	[49]
*Decapentaplegic (dpp)*	Drosophila	*BMP2/BMP4*	*Grh* binds to a *dpp* repression element in a ventral repression region	[49]
*Proliferating cell nuclear antigen (Pcna)*	Drosophila	*PCNA*	*Grh* binds to *PCNA* to regulate cellular differentiation	[50]
*Castor (Cas)*	Drosophila	*CASZ1*	*Grh* may be the final step of a signalling cascade involving *Cas* that regulates the differentiation of embryonic neuroblasts	[51]
*Suppressor of Hairless (Su(H))*	Drosophila	*RBPJ*	*Su (H)* and *Grh* may regulate both *kni* and *knrl* in regulating dorso-ventral patterning of the primitive gut tube.	[52]
*Knirps (kni)*		*RAR related orphan receptor A*		
*Knirps-related (knrl)*		*NR1H2*		
*Krotzkopf verkehrt (kkv)*	Drosophila	*HAS1/2/3*	These genes, when inactivated in Drosophila, lead to a “blimp” phenotype following removal of the vitelline membrane of the eggshell.	[53]
*Knickkopf (knk),*		*THBD*		
*Retroactive (rtv),*		No known orthologues	Phenotypically similar to *Grh* mutants.	
*Zeppelin (zep) [renamed Gfat1]*		*GFPT1/2*		
*Fasciclin II (FasII)*	Drosophila	*NCAM1/2*	Compound mutants of *Grh*, *FasII* and/or *ATPa* show additive defects in epithelial integrity/cohesion.	[54]
*ATPase a sub-unit (ATPa)*		*ATPA* family		
*Branchless (Bnl; upstream)*	Drosophila	*FGF* family	An upstream signalling factor that upregulates *Grhl* expression in Drosophila airway development	[55]
*Ribbon (rib)*	Drosophila	*BTBD18/ZBTB9/ZBTB12*	Hypothesised that interaction between *Grh* and *Rib* may regulate tracheal branching and apical membrane growth; possibly via *Bnl* signalling.	[54]
*Mab-5 (Ubx orthologue)*	C. elegans	*HoxB6*	Promoter binding analyses show that *Grh* can bind to gene promoters, as in Drosophila, in post-embryonic development	[4]
*dbl-1 (dpp orthologue)*		*BMP* Family		
*Aromatic L-aa decarboxylase (Ddc orthologue)*		*DDC*		
*Starry Night/Flamingo (stan/flam)*	Drosophila	*Celsr1*-*3*	*Grh* regulates *stan/flam* levels in Drosophila wings during development.	[55]
*Frizzled (Fz)*		*Frizzled*	*Fz* is an upstream factor of *Grh*, and *Grh* is crucial for *Fz* function in apical accumulation of *Fz*. Interacts genetically with *Grh* during wing development.	
*Optomotor-blind (Omb)*		*Tbx2*	Predicted *Grh* target (*in silico*). Mutant shows defects in cuticle pattern formation, cell shape, pigmentation and hair-organisation. Slight interaction with *Grh* in wing development.	
*Ovo/Shaven Baby(ovo/svb)*		*Ovol1*	Predicted *Grh* target (*in silico*). Mutant shows imaginal disc morphogenesis defects; no interaction with *Grh* observed.	
*Scabrous (sca)*		partial homology to *fibrinogen*	Predicted *Grh* target (*in silico*). Mutant shows imaginal disc morphogenesis defects; no interaction with *Grh* observed.	
*Pipsqueak (psq)*		*BACH2/BTBD18/DNAJC14*	Predicted *Grh* target (*in silico*). Mutant shows defects in early egg and embryo patterning; no interaction with *Grh* observed.	
*Ventral veins*		*POU3F2*	Predicted *Grh* target (*in silico*). Binds to a DNA sequence element required for expression of the *Ddc* gene. Could play a role in early ectodermal cells; no interaction with *Grh* observed.	
*lacking/Drifter (vvl)*				
*Abdominal A (Abd A)*	Drosophila	*HoxA6/B6/C6*	*Grh* is required to maintain *AbdA* expression in neuroblasts in late larval phase, although it is not required for initial *AbdA* initiation in these cells.	[56]
*Pale (ple)*	Drosophila	*Tyrosine Hydroxylase*	*Ple* contains 3 *Grh*-binding sites (as well as others) in 2 enhancer regions within the promoter. Potential role of *Grh*-*Ple* in wound repair &/or cross-linking of epidermal genes.	[46]
*E-Cadherin (shotgun)*	Drosophila	*E-Cadherin*	*E-Cadherin* is greatly reduced in *Grh*-deficient post-embryonic neuroblasts, ectopic *Grh* expression increases *E-Cadherin* expression, and the *E-Cadherin* promoter contains binding sites for *Grh*.	[57]
*Pleiohomeotic (Pho)*	Drosophila	*ZFP42*	*Grh* may be responsible for targeting this Polycomb group (*PcG*) gene to its specific binding sites in target promoters. Polycomb genes are involved in silencing genes that control development	[58]
*Thelytoky (th)*	A. mellifera (Honeybee)	*Grhl* family	The *th* gene may be responsible for regulating reproduction and sterility	[59]
*FasciclinIII (FasIII)*	Drosophila	*Nectin3*	Flies over-expressing *Grh* show strong up-regulation of *FasIII*, coracle and sinuous and these pathways contribute to the formation of apical barriers in the epidermis.	[60]
*Coracle (cor)*		*EPB41L3*		
*Sinuous (sinu)*		*Claudin* Family		
*Neurexin (nxr)*		*Neurexin* Family	Potential role for interaction of these genes with *Grh* in the formation of septate junctions in flies (analogous to adherens junctions in mammals).	
*Discs-large (Dlg)*		*Dlg* Family		
*Tailless (tll)*	Drosophila	*NR2E1*	Ras-signalling, possibly involving removal of *Grh* repressors, can transactivate the expression of these two genes. *Grh* can also bind tll regulatory regions.	[61]
*Hueckebein (hkb)*		*KLF1/14/16*		
*Intermediate Neuroblasts Defective (Ind)*	Drosophila	*GSX1*/*GSX2*	*Grh* binds to *ind* to putatively regulate dorso-ventral patterning	[62]
*Flotillin-2 (flot-2)*	Drosophila	*Flot*-2	*Grh* binding sites are present on *flot-2* and may contribute to epidermal wound repair.	[63]

**Table 2 ijms-23-02735-t002:** Known or hypothesised direct target genes of the *Grhl* family characterised in non-mammalian vertebrate species, namely the African clawed-frog (*Xenopus laevis*), and the Zebrafish (*Danio rerio*).

GENE(S)	SPECIES	BIOLOGICAL ROLE	REFERENCE(S)
*Cytokeratin 1 (XK81A1)*	Xenopus	*xGrhl1* down-regulation results in downregulation of *XK8IAI* with concomitant inhibition of terminal epidermal differentiation.	[64]
*Transcription factor AP2 (AP-2)*		*AP*-2 co-operates with *xGrhl1* to transactivate the *xKI81A* reporter.	
*Distal-less Homeobox 3 (Dlx3)*		*Dlx3* and *Dlx5* may be induced by *xGrhl1* over-expression.	
*Distal-less Homeobox 5 (Dlx5)*			
*Claudin B (cldnb)*	Zebrafish	*Grhl2b* binds to claudin B and epcam in the otic vesicles in zebrafish. Loss of *grhl2b* leads to loss of both genes, and manifests as loss of balance and hearing behaviour in zebrafish.	[6]
*epithelial cell adhesion molecule (epCAM)*			
*Engrailed 2a (eng2a)*	Zebrafish	*Grhl2b* mediated regulation of eng2a drives patterning and survival of neuroblasts at the midbrain-hindbrain boundary.	[13]
*Small protein effector of cdc42 1 (spec1)*		Bound by *grhl2b* to drive the morphogenesis of the midbrain-hindbrain boundary.	
*Endothelin 1 (edn1)*	Zebrafish	*Grhl3* regulation of *Edn1* is critical for maintaining neural crest cell (NCC) fidelity in the branchial arches, and consequently, for lower jaw development.	[7]
*IRF6*	Zebrafish/Xenopus (upstream regulator)	*Irf6* binds to the *grhl3* promoter and drives periderm differentiation. *Grhl3* rescues *Irf6* deficiency in both zebrafish and Xenopus	[65]
*Rho Guanine Nucleotide Exchange Factor 19 (arhgef19)*	Zebrafish	*grhl3* regulates *arhgef19* in the morphogenesis of the midbrain-hindbrain boundary.	[66]

**Table 3 ijms-23-02735-t003:** Known or hypothesised direct target genes of the *Grhl* family characterised in rodent models, rodent-derived cell lines, and cell lines derived from other mammalian models.

GENE(S)	SPECIES	BIOLOGICAL ROLE	REFERENCE(S)
*LIM Domain Only 4 (Lmo4)*	Mouse	*Lmo4* is co-expressed in several of the same regions as *Grhl3* during mouse development, esp. the ectoderm, oral and dental epithelium. *Lmo4* also dimerizes with *Grhl3*.	[67]
*Protein Kinase C isoforms (PKCa, bI, bII, d, e, g)*	Mouse (Upstream regulators)	Involved in mediating the inositol-responsive rescue of neural-tube defects in *curly-tail* mutants. Action is potentially via stimulating cell proliferation (particularly in hindgut).	[68]
*Transglutaminase 1 (Tgm1)*	Mouse	Involved in epithelialisation through cross-linking barrier components.	[20]
*Tumor protein 53 (Tp53)*	PC12 cells (Rat)	Activation of *p53* by NGF during neurite differentiation results in up-regulated *Grhl3* expression, and *p53* binds to the *Grhl3* promoter.	[69]
*Desmoglein1 (Dsg1)*	Mouse	*Dsg1* is a direct target of *Grhl1*, but not *Grhl3*. Involved in mediating hair anchorage in epidermis through regulation of desmosomes.	[70]
*Transforming growth factor alpha (TGF-a)*	Mouse	*Grhl3* acts upstream of *TGF*-a and is required for the formation of the eyelids in mouse. No evidence of direct binding.	[71]
*Uroplakin II (Upk2)*	Mouse	Regulates differentiation of umbrella cells and urinary bladder barrier formation.	[72]
*E-cadherin (Cdh1)*	Mouse	*Grhl2* is responsible for directly regulating the expression of these genes in various epithelial tissues. Downregulated in *Grhl2*^−/−^ collecting ducts.	[15,73,74]
*Claudin 4 (Cldn4)*			
*Rho Guanine Nucleotide Exchange Factor 19 (Arhgef19)*	Mouse	*Arhgef19* is bound by *Grhl3* and acts as part of the Planar Cell Polarity (PCP) pathway in keratinocytes.	[75]
*Phosphatase and tensin homolog (Pten)*	Mouse	*Pten* acts downstream of *Grhl3* to suppress squamous cell carcinomas.	[26]
*Lamin B1*	Mouse	*Lamin* expression modifies the spina bifida presentation in the *grhl3*-hypomorphic model *curly tail*. No direct binding demonstrated.	[76]
*Homeobox Protein Nkx2-1*	Mouse	*Grhl2* and *Nkx2*-1 form a reciprocal regulatory loop in mouse lung epithelium.	[77]
*Forkhead box M1b (FoxM1b)*	Mouse	*Grhl2* induces *FoxM1B* downstream of HPV-16 in the context of oral cancer in human oral keratinocytes.	[78]
*micro-RNA-21 (miR-21)*	Mouse	Decreased levels of *Grhl3* contributed to squamous cell carcinoma via an upregulation of *miR-21*, contributing to a feedback loop involving *Msh2*.	[29]
*Erb-B2 Receptor Tyrosine Kinase 3 (erbb3)*	Mouse	*Grhl2*-mediated direct regulation of *Erbb3* regulates proliferation, EMT and morphology in breast cancer cell lines.	[32]
*Member RAS Oncogene Family 25 (Rab25)*	Mouse	*Grhl2*-mediated regulation of *Rab25* promotes localisation of *Cldn4* at tight junction in cholangiocytes. Regulates lumen expansion and barrier formation in kidney epithelia. Downregulated in *Grhl2*^−/−^ collecting ducts.	[73,74,79]
*Histone deacetylase 3 (HDAC3)*	Mouse	*HDAC3* and *MYCN* directly inhibit *Grhl1*, therein promoting tumourigenesis in neuroblastoma cell lines.	[39]
*Myc Family Member (MYCN)*			
*Calcium channel, voltage-dependent, L type, alpha 1D subunit (Cacna1d)*	Mouse	*Cacnd1* and *Xpo1* are predicted targets based on the presence of a conserved *Grhl*-binding site in their promoters; *Ddc* and *Flot2* are orthologues of known *Drosophila grh* targets. Expression of all was reduced in the kidney of *Grhl1*^−/−^ mice. Direct binding not shown.	[80]
*Ddc (Dopa decarboxylase)*			
*Flotillin 2 (Flot2)*			
*Exportin 1 (Xpo1)*			
*Ovo-like 2 transcription factor 2 (Ovol2)*	Mouse	*Grhl2* positively regulates *Ovol2*, and *Ovol2* can rescue lumen defects in *Grhl2*^−/−^ mice, via regulation of *Cldn4* and *Rab25*. Downregulated in *Grhl2*^−/−^ collecting ducts.	[73,74]
*Potassium Two Pore Domain Channel Subfamily K Member 1 (Kcnk1)*	Mouse	*Grhl2* target genes involved in placental development in mice, as verified by direct promoter binding (ChIP).	[81,82]
*Ladinin 1 (Lad1)*			
*Leucine Zipper Down-Regulated In Cancer 1 (Ldoc1)*			
*Placenta-enriched 1 (Plac1)*			
*Prominin 2 (Prom2)*			
*Member RAS Oncogene Family 15 (Rab15)*			
*Sphingomyelin Phosphodiesterase Acid Like 3B (Smpdl3b)*			
*Serine Peptidase Inhibitor, Kunitz Type 1 (Spint1)*		Additionally, *Grhl2* driving *Spint1* may play a role in branching morphogenesis and development of the salivary gland.	
*Testis Expressed 19 (Tex19.1)*			
*Transcription Factor AP-2 Alpha (Tfap2a)*			
*Transmembrane Protein 54 (Tmem54)*			
*Tyrosine Kinase Non Receptor 1 (Tnk1)*			
*1600014K23Rik (possibly ferritin, heavy polypeptide-like 17, member A; Fthl17a)*			
*Glycogen synthase kinase 3 (Gsk3b)*	Mouse (conditional deletion model)	Direct binding validated by ChIP and EMSA in oral epithelial cells. *Grhl3*-*GSK3b* pathway regulates proliferation in both normal and oral-epithelial cancer cells.	[27]
*p300*	Dog (MDCK cells)	*Grhl2*-mediated suppression of *p300* led to reduced induction of matrix metalloproteases and suppression of tubulogenesis. No direct binding shown.	[36]
*Zinc Finger E-Box Binding Homeobox 1 (ZEB1)*	Mouse	*Grhl2* and *Zeb1* form a double negative regulatory feedback loop in breast cancer cell lines.	[32,38]
*Myosin Heavy Chain 10 (Myh10)*	Mouse	These genes were down-regulated in kidney collecting ducts lacking *Grhl2*, and were confirmed as direct binding targets via ChIP.	[74]
*Shroom Family Member 3 (Shroom3)*			
*Tight Junction Protein 3 (Tjp3)*			
*E74 Like ETS Transcription Factor 5 (Elf5)*	Mouse	*Grhl2* binds to the *Elf5* promoter in whole embryonic lung tissue, and this pathway may regulate proliferation in distal progenitor cells.	[19]
*Fascin Actin-Bundling Protein 1 (Fscn1)*	Mouse	*Grhl3* regulates *Fscn1* in keratinocytes during wound healing.	[83]

**Table 4 ijms-23-02735-t004:** Known or hypothesised direct target genes of the *GRHL* family characterised in human tissue, human cancer cell lines or other cell lines of human origin.

GENE(S)	SPECIES	BIOLOGICAL ROLE	REFERENCE(S)
*Engrailed 1 (En1)*	Human	*Grhl1* binds to and transactivates promoter of human En1 in vitro.	[2]
*Tumour necrosis factor a (TNF-a)*	MCF-7 cells (Human; upstream regulator)	*TNF*-*a* induces *Grhl3* expression in *MCF* tumour cells. Hypothesised to contribute to endothelial cell migration and angiogenesis	[84]
*Human telomerase reverse transcriptase (hTERT)*	SCC4 cells – Human	*Grhl2* knockdown has inhibitory effect on telomerase activity, thus potentially implicating it in tumourogenesis.	[85]
*Protein Kinase Akt; endothelial Nitric Oxide Synthase (eNOS)*	Human	*akt* and *eNOS* are induced by *Grhl3* in endothelial cells; no direct binding demonstrated.	[86]
*Zinc Finger Enhancer-Binding Protein 1 (Zeb1)*	Human	*Grhl2* directly repressed transcription of *Zeb1* in a breast cancer cell line, resulting in reduced EMT, enhanced anoikis sensitivity and upregulation of *Smad*, *mir200b*/c and *BMP2*.	[10]
*Claudin 3 (Cldn3)*	Human	Regulation of tight junctions in the epithelial barrier relies on *Grhl2* expression and downstream activation of *claudin3*, *4*. No direct binding demonstrated.	[87]
*Cyclin B2 (CCNB2)*	SCC4 cells–Human	*Grhl2* binds to the proximal promoter regions of these target genes in Squamous Cell Carcinoma (SCC) cell lines, and influences histone modification at the target gene promoters.	[78]
*Proliferating Cell Nuclear Antigen (PCNA)*			
*Ki-67*			
*Small Proline Rich Protein 2a (Sprr2a)*			
*B-cell lymphoma 2 (BCL2)*	Human	Overexpression of *Grhl2* decreased *c-Myc* and *Bcl-2* expression in SGC7901 gastric cancer cells. No direct binding demonstrated.	[88]
*Myc Family Member (c-Myc)*			
*Multiple putative target genes (n = 296)*	Human	Multiple targets predicted by cross-referencing ChIP-SEQ data and RNA-SEQ data from human bronchial epithelial (HBE) cells expressing dominant-negative *Grhl2*	[11]
*Interferon Regulatory Factor 6 (IRF6; upstream)*	Human	*IRF6* binds to an enhancer element near the *Grhl3* promoter, and may regulate the development of the periderm in both zebrafish and human keratinocytes. Mutations in *Grhl3* and *IRF6* are also both causative of the human palatal clefting condition Van der Woude Syndrome.	[12,65]
*Micro-RNA-122 (miR-122)*	Human	*Grhl2* negatively regulates *miR-122* in hepatocyte differentiation.	[89,90]
*Small Cell Adhesion Glycoprotein (SMAGP)*		*Grhl2*-mediated regulation of *SMAGP* and *ZNF50* genes is predicted to regulate airway cell polarity, maintain barriers and cell differentiation.	
*Zinc Finger Protein 750 (ZNF750)*		The *Grhl2*-*ZNF50* pathway may regulate ciliogenesis.	
*Glycogen synthase kinase 3 (Gsk3b)*	Oral epithelial and cancer cell lines, Human	Direct binding validated by ChIP and EMSA in oral epithelial cells. *Grhl3-GSK3b* pathway regulates proliferation in both normal and oral-epithelial cancer cells.	[27]
*Tumor protein 63 (p63)*	Human	*Grhl2* and *p63* form a regulatory feedback loop which governs epithelial traits in human keratinocytes.	[91]
*Micro-RNA-217 (miR-217; upstream)*	Human	*miR-217* inhibits keratinocyte proliferation whilst promoting differentiation.	[92]
*Micro-RNA-200 (miR-200)*	Human	*Grhl2* binds in the promoter region of *miR-200*; *Grhl2* may regulate the promoter of a polycistronic primary transcript	[38]
*P300*	Human	A small sequence of *Grhl2* between aa 425-437 is responsible for inhibiting the C-terminal Domain of *P300* and was postulated to contribute to the suppression of EMT and tubulogenesis.	[36]
*Tetraspannin CD9 (CD9)*	Human	*CD9* is a direct target of *Grhl1*, and high *CD9* expression was associated with favourable outcomes in neuroblastomas.	[39]
*Octamer-binding transcription factor 4, (OCT-4)*	Human	*Grhl2* binds to the proximal region of the *Oct-4* promotor, putatively regulating the transcription of *Oct-4* in cell reprogramming and contributing to oral cell carcinoma.	[93]
*Matrix Metalloproteinase 2, 7 and 9 (MMP 2/7/9)*	Human	*GRHL2* expression correlated with expression levels of these three *MMPs* in gastric cancer cells; no direct binding demonstrated.	[94]
*Micro-RNA-194 (miR-194; upstream)*	Human	Luciferase assays showed that *miR-194* binds to the 3’UTR region of *Grhl2*.	[95]
*Ras Homolog Family Member G (RhoG)*	Human	RhoG possesses 3 *GRHL2* binding sites, and *Grhl2* inhibits RhoG expression	[96]
*Micro-RNA-133a (miR-133a; upstream)*	Human	*miR-133a* was demonstrated by Luciferase assay as binding to *Grhl2* and subsequently regulating *ESRP* expression and EMT in the airway epithelium	[97]
*Lysine Methyltransferase 2C (KMT2C)*	Human	*Grhl2* recruits *KMTC* and *KMTD* to interact with *Grhl2*-induced enhancers	[98]
*Lysine Methyltransferase 2D (KMT2D)*			
*Nitric Oxide Synthase 3 (NOS3)*	Human	*Grhl3* and *eNOS* co localise as observed by proximity ligation assays, however no direct demonstration of *Grhl3* regulation of *eNOS*.	[99]
*Extracellular signal-regulated kinase (ERK; upstream)*	Human	*Erk* phosphorylates *Grhl1*, leading to translocation of *Grhl1* to the nucleus and contributing to cell cycle proliferation in cancer cells.	[100]
*S100 Calcium Binding Protein A10 (S100A10)*	Human	*Grhl2* binds to a site in the first intron of *S100A10* and drives transcription in HEK-293 cells, and this pathway may drive extracellular matrix degradation in cancer.	[101]
*Cell Division Cycle 27 (CDC27)*	Human	*GRHL1* binds to the promoters of these genes and directly regulates their expression in non-small cell lung cancer (NSCLC) cells.	[100]
*RAD21 cohesive complex (RAD21)*			
*Cell division cycle 7 (CDC7)*			
*Anaphase promoting complex subunit 13 (ANAPC13)*			

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
