# Peer review of "Grainyhead-like (Grhl) Target Genes in Development and Cancer"

_ijms, 2022, doi:10.3390/ijms23052735_

Round 1

Reviewer 1 Report

In the manuscript ''Grainyhead-like (Grhl) target genes in development and cancer'' authors present recent knowledge on the grainyhead-like (GRHL) factors, transcription factors that are pivotal during embryogenesis as well as in (de)regulation of growth and survival pathways in cancer. Their focus of interest is animal and in vitro models (xenopus, zebrafish, drosophila, mouse, human) which pose the basis of their understanding of three GRHL family members that they further elaborate through their role in development and cancer of different organs (brain, epidermis, head and neck, kidney, breast, lung, liver, gastrointestinal track and bladder, endothelium, placenta, bone).

Broad comments: The manuscript is a pleasure to read and it brings insight into an important matter, however there are some small issues that have to be raised to improve the manuscript

Specific comments:

  1. Tables are difficult to read (resolution, font?), and there are no names on the tables (Table 1,2,3,4).
  2. References are not appropriately cited.
  3. It would be of interest for the readership to include at least one figure in which the authors would summarize their knowledge/ proposed mechanism of grh/grhl pathways eg. Grh targets (cdh1, eng2a, spec1) and upstream regulators (fgf8…)

Author Response

Reviewer #1

In the manuscript ''Grainyhead-like (Grhl) target genes in development and cancer'' authors present recent knowledge on the grainyhead-like (GRHL) factors, transcription factors that are pivotal during embryogenesis as well as in (de)regulation of growth and survival pathways in cancer. Their focus of interest is animal and in vitro models (xenopus, zebrafish, drosophila, mouse, human) which pose the basis of their understanding of three GRHL family members that they further elaborate through their role in development and cancer of different organs (brain, epidermis, head and neck, kidney, breast, lung, liver, gastrointestinal track and bladder, endothelium, placenta, bone).

Broad comments: The manuscript is a pleasure to read and it brings insight into an important matter, however there are some small issues that have to be raised to improve the manuscript

Specific comments:

  1. Tables are difficult to read (resolution, font?), and there are no names on the tables (Table 1,2,3,4).

We have improved the clarity and formatting of our tables, and have now included them as separate word documents (also provided in response to reviewer #2). The clarity should now be substantially improved; we have included table numbers as part of the figure legends for each table.

  1. References are not appropriately cited.

We have now corrected the formatting to the approved IJMS style.

  1. It would be of interest for the readership to include at least one figure in which the authors would summarize their knowledge/ proposed mechanism of grh/grhl pathways eg. Grh targets (cdh1, eng2a, spec1) and upstream regulators (fgf8…)

We felt this was an excellent suggestion, and have in fact undertaken a substantial effort to now include four separate figures to highlight experimentally-validated Grhl-relationships. Our figures identify the organs and tissue types in which Grhl-pathways have been identified in each of Drosophila, mouse, human and zebrafish. We believe that these have substantially improved our manuscript (also provided in response to reviewer #2).

Reviewer 2 Report

The review by Gasperoni et al., entitled “Grainyhead-like (Grhl) target genes in development and cancer” deals with a topic of general interest and fits perfectly with the topic of the special issue Conserved Pathways in Development and Cancer. 

General comments:

The article is very dense and well documented, but it is also difficult to read due to the lack of figures to support the different points made. The molecular function of these factors as epithelial and pioneer transcription factors could be better emphasised and elaborated upon. What does "integral transcription factor" mean?

Furthermore, the review focuses more on the function of Grh-like factors in development and cancer than on their target genes. Two options are possible:

1)- change the title and remove target genes

2)- improve the notion of target genes as the molecular aspect is not addressed and it is difficult to understand which genes are direct or putative targets. For example: page 8: Whole genome analysis of neuroblastoma cells identified 170 genes as potential targets of GRHL1. Page 11: Relevant GRHL2 target genes may include...

In this line, the tables are not well set up. It is difficult to know which target genes are common to all species. Are these genes direct or indirect and tissue-specific targets?

Minor comments:

- The tables are not well drawn.

- The font changes in some paragraphs and probably highlights the different authors. It is often difficult to follow the idea during these transitions.

- As the authors have mentioned several models/organs, the reader needs some context to understand them. Example page 13: GRHL2 is expressed in p53+ and KRT5+ basal cells. I am not sure what the authors are talking about. Illustrations may help.

- Paragraphs 11-14 could easily be deleted to make room for the illustrations.

Author Response

Reviewer #2

The review by Gasperoni et al., entitled “Grainyhead-like (Grhl) target genes in development and cancer” deals with a topic of general interest and fits perfectly with the topic of the special issue Conserved Pathways in Development and Cancer. 

General comments:

The article is very dense and well documented, but it is also difficult to read due to the lack of figures to support the different points made.

We agree, and have now added four new figures to visually represent experimentally-validated Grhl relationships.

The molecular function of these factors as epithelial and pioneer transcription factors could be better emphasised and elaborated upon.

The reviewer is certainly correct that the grh/Grhl family are pioneer TFs, and in fact this has been described in several publications (e.g. Jacobs, Nat Gen, 2018; Nevil, Development 2020; Sundarajan et al., FMB 2020). However, the focus of our review is not so much on grh/Grhl function, as it is on reviewing the literature pertaining to grh/Grhl interactions with target genes.

Having said that, we do recognise that this role of grh/Grhl factors is important to highlight, and so we have now included the following text within the introduction explaining that grh/Grhl proteins do indeed function as pioneer TFs (lines 78-82):

“Moreover, the grh/Grhl factors have recently been characterised as “pioneer TFs” [22-24], that is, they are able to access DNA-binding sites within typically inaccessible (closed) chromatin, to facilitate direct target gene transcription, further solidifying the essential nature of these TFs in normal cellular function and as key drivers of embryogenesis.”

 What does "integral transcription factor" mean?

In this context “integral” only refers to the fact that the grh/Grhl TFs are crucial, and in fact necessary, for normal development, rather than ascribing any particular defined terminology to their function. However, to avoid confusion we have changed the text in the abstract to replace “integral” with “essential, highly conserved”

Furthermore, the review focuses more on the function of Grh-like factors in development and cancer than on their target genes.

As stated, the purpose of our review is not necessarily to review the specific functions of grh/Grhl factors and their targets across disparate contexts in substantial detail, but rather, to highlight and identify genetic relationships that have been experimentally validated in the literature. The idea here is that this review will serve as a foundational resource for future studies to address the role of certain genes in development/cancer within a particular organ or tissue. As the reviewer notes, our manuscript is already quite extensive, and we feel that expanding upon these genetic relationships in significant detail would make it an unwieldy publication.

Two options are possible:

1)- change the title and remove target genes

2)- improve the notion of target genes as the molecular aspect is not addressed and it is difficult to understand which genes are direct or putative targets.

This is a good point. Unfortunately in many cases we are constrained by the published data, which although often discusses a functional relationship, does not provide ChIP or other direct binding assay data to determine whether the binding is direct or indirect. However, we feel that it is important to not exclude these relationships from our analysis, as they are clearly “biologically-relevant”. It is of course possible that grh/Grhl may heterodimerise with other protein partners to drive transcription of these genes, further complicating the simple notion of “direct/indirect” transactivation.

Where possible in our tables we have explicitly stated if no direct-binding data exist, which should serve to eliminate some confusion.

For example: page 8: Whole genome analysis of neuroblastoma cells identified 170 genes as potential targets of GRHL1. Page 11: Relevant GRHL2 target genes may include...

We agree that not all studies we review in our manuscript clearly define the functional nature of grh/Grhl interaction with their target genes; the neuroblastoma paper identified by the reviewer being a case in point. However, we feel that including these studies within our review is essential, as it allows future experimentally-determined target genes to be “linked” to previously-identified putative, or predicted targets. By presenting the current state of knowledge regarding both direct and “unknown” targets in the one publication, it will be an invaluable source of information others in the field of grh/Grhl research.

In this line, the tables are not well set up. It is difficult to know which target genes are common to all species.

This is a very good point. Unfortunately, typically once a gene has been identified as a direct target in one species, no further work is done in other species to determine whether the relationship is conserved (save for a few key genes such as cdh1, eng2a, cldn4 etc.). Therefore, we felt it would be misleading to only highlight these few genetic interactions as being “conserved across species”, as it could suggest that other genetic interactions are not conserved, which is incorrect. It is extremely likely (if not certain) that many, many more of these functional relationships operate across disparate species than what has been experimentally validated to date.

Are these genes direct or indirect and tissue-specific targets?

All genes listed in the table are direct targets, unless otherwise specifically noted. By “direct targets”, we mean that there is evidence of functional regulation (e.g. through specific activation assays in vitro or in vivo, or changes in gene expression following grh/Grhl dysregulation), together with clear evidence that grh/Grhl binds to the gene promoter (e.g. via ChIP or ChIP-Seq assays). We have included this clarification in our manuscript (lines 112-116).

Minor comments:

- The tables are not well drawn.

As provided in response to reviewer #1, we have improved the clarity and presentation of our tables, and can make further changes if deemed necessary at the editing stage.

- The font changes in some paragraphs and probably highlights the different authors. It is often difficult to follow the idea during these transitions.

We are unsure why this was the case, but in any case we have now re-formatted the manuscript to ensure consistency in font, as well as to amend the referencing style to be appropriate for IJMS. We believe that transitions between paragraphs are easy to follow, however we would happily clear up any paragraphs where this is not the case.

- As the authors have mentioned several models/organs, the reader needs some context to understand them. Example page 13: GRHL2 is expressed in p53+ and KRT5+ basal cells. I am not sure what the authors are talking about. Illustrations may help.

We have clarified that these “basal cells” are “airway epithelial progenitor cells” (line 544). We believe that all other cells, organs and tissue types are adequately defined in the text, but we will gladly make further amendments if required.

- Paragraphs 11-14 could easily be deleted to make room for the illustrations.

We would be comfortable in removing these paragraphs if deemed essential, although these have been included to highlight that separate models with abrogated Grhl function present with differing neurulation phenotypes. The reason for this is unclear, however the most likely explanation is that differing Grhl-mutations result in differentially-penetrant loss of Grhl-function, leading to altered activation of target genes. We feel this is an important point to remember when discussing Grhl-biology.

Round 2

Reviewer 2 Report

The authors have taken into account the vast majority of my requests. 
Congratulations to them.